Run for your life: can exercise be used to effectively target GLUT4 in diabetic cardiac disease?

Bowman Peter R.T. 1
http://orcid.org/0000-0003-4821-9741 Smith Godfrey L. 2
http://orcid.org/0000-0001-6571-2875 Gould Gwyn W. 1 gwyn.gould@strath.ac.uk
1 Strathclyde Institute of Pharmacy and Biomedical Sciences, University of Strathclyde , Glasgow , United Kingdom
2 Institute of Cardiovascular and Medical Sciences, University of Glasgow , Glasgow , United Kingdom
McClelland Grant
Electronic publication date: 2021 May 25
Publication date: 2021
Volume: 9
Electronic Location ID: e11485
Received 2021 Jan 10; Accepted 2021 Apr 27
Copyright: © 2021 Bowman et al.
Copyright year: 2021
Copyright holder: Bowman et al.
License: This is an open access article distributed under the terms of the Creative Commons Attribution License, which permits unrestricted use, distribution, reproduction and adaptation in any medium and for any purpose provided that it is properly attributed. For attribution, the original author(s), title, publication source (PeerJ) and either DOI or URL of the article must be cited.
License URL: https://creativecommons.org/licenses/by/4.0/

Keywords: GLUT4, Diabetic cardiomyopathy, Cardiomyocyte, Exercise, Glucose

Funding: British Heart Foundation PG/18/47/33833 PhD scholarship FS/14/61/31284 Work in the GLS and GWG groups is supported by British Heart Foundation project grant PG/18/47/33833 (Godfrey L. Smith and Gwyn W. Gould) and PhD scholarship FS/14/61/31284 (to Peter R.T. Bowman). The funders had no role in study design, data collection and analysis, decision to publish, or preparation of the manuscript.

==============================
The global incidence, associated mortality rates and economic burden of diabetes are now such that it is considered one of the most pressing worldwide public health challenges. Considerable research is now devoted to better understanding the mechanisms underlying the onset and progression of this disease, with an ultimate aim of improving the array of available preventive and therapeutic interventions. One area of particular unmet clinical need is the significantly elevated rate of cardiomyopathy in diabetic patients, which in part contributes to cardiovascular disease being the primary cause of premature death in this population. This review will first consider the role of metabolism and more specifically the insulin sensitive glucose transporter GLUT4 in diabetic cardiac disease, before addressing how we may use exercise to intervene in order to beneficially impact key functional clinical outcomes.

Introduction—identifying the heart of the issue

Diabetes is defined clinically by a fasting blood glucose value of >7 mM, or fasting HbA1c level >6.5%, predominantly due to peripheral insulin resistance (type 2 diabetes; T2D) or insufficient pancreatic insulin production (type 1 diabetes; T1D). The primary consequence of chronic hyperglycaemia is vascular degeneration (Dal Canto et al., 2019), which accounts for most common complications of diabetes e.g., diabetic retinopathy, diabetic foot and myocardial infarction (MI). Resulting increased blood pressure and coronary artery disease may also overload the heart and lead to the development of cardiac failure.

In 1972 a post-mortem of four diabetic patients identified cardiac failure in the absence of vascular disease, indicating for the first time that diabetes may directly impair cardiac structure and function (Rubler et al., 1972). This phenomenon is now referred to as diabetic cardiomyopathy (DCM), which first presents as an impairment in diastolic function early after the onset of diabetes. Early diastolic dysfunction has been repeatedly demonstrated in cohorts of asymptomatic diabetic patients using several echocardiographic parameters, primarily related to transmitral blood flow and myocardial tissue movement velocities during early and late diastole (Nicolino et al., 1995; Di Bonito et al., 1996; Poirier et al., 2001; Poulsen et al., 2010). Impaired relaxation of the ventricular wall is likely to delay early diastolic filling and therefore increase the contribution of atrial contraction mediated blood flow to ensure sufficient ventricular filling. If left unabated, major structural changes and systolic dysfunction develop and drive progression of DCM towards heart failure (Bell, 2003).

The estimated prevalence of DCM varies, largely due to the sensitivity of the detection system and criteria utilised. Boyer et al. demonstrated that within a population of asymptomatic diabetics conventional Doppler echocardiography detected diastolic dysfunction in 46% of individuals, whereas this increased to 63% using tissue Doppler imaging and 75% with a combined approach (Boyer et al., 2004). The high prevalence and potentially severe life limiting consequences of this condition coupled with a lack of specific treatment options makes this an area of urgent unmet clinical need.

The pathophysiological mechanisms associated with DCM are complex, both in isolation and in the way that they interact with and amplify one another via a myriad of signalling pathways (Jia, Hill & Sowers, 2018). Factors including activation of the renin-angiotensin and sympathetic nervous systems, inflammation, oxidative and endoplasmic reticulum stress and increased production of advanced glycation end products (AGEs) have all been cited (Fig. 1). Importantly, many of these are secondary factors. While important contributors to disease progression, they are probably consequences of earlier events. These issues are worthy of review in their own right; however, here we will focus on key early instigators of DCM that arguably have the most potentially powerful disease modifying capacity e.g., metabolic changes linked to the glucose transporter GLUT4. We will then consider whether lessons from exercise physiology could offer a useful mechanism for treatment of DCM. This review is aimed at those with an interest in cardiovascular disease, exercise physiology and the metabolic basis of heart disease

Figure 1 Mechanisms of diabetic cardiomyopathy.

A range of contributory factors in the development of diabetic cardiomyopathy have been proposed. These include lipotoxicity, impaired insulin signalling, accumulation of advanced glycation end products, ER and oxidative stress. We focus on metabolic disturbance, specifically defects in the GLUT4 glucose transport system.

Survey methodology

We conducted a literature search using combinations of key words (e.g., GLUT4 and diabetic cardiomyopathy) and ad hoc searches as required, using PubMed, MedLine or Web of Science providing an objective subject coverage. We selected articles in English and based on our subjective valuation of the work. This is therefore a comprehensive but not exhaustive review.

RESULTS

The pathophysiology of diabetic cardiomyopathy

Cardiac structure and function are intimately linked. A key facet of DCM is the onset of pathological left ventricular hypertrophy; however, it is important to note that often diastolic dysfunction is detected in diabetic human patients with no corresponding structural abnormalities (Schannwell et al., 2002). Additionally, whilst systemic hyperglycaemia is a fundamental feature of DCM (and diabetes in general), evidence from rodent models of diabetes indicates that cardiac specific insulin resistance is one of the earliest detectable pathological events, prior to the deterioration of cardiac function (Park et al., 2005). This suggests that intrinsic alterations within the myocardium may initiate DCM prior to manifestation of systemic insulin resistance. This may therefore arise before external (consequential or compensatory) factors drive disease progression. At a functional level, this points towards deficits in cardiomyocyte excitation contraction coupling [for a detailed review of this process see (Bers, 2002)]. Indeed, evidence from rodent models of DCM has established impaired cardiomyocyte calcium handling as a primary limitation (Belke & Dillmann, 2004; Pereira et al., 2006; Stølen et al., 2009). Studies of T2D patients have hinted at similar deficits (Lamberts et al., 2014; Daniels et al., 2015).

Cardiac excitation contraction coupling is an energy intensive process dependent upon ATP availability. Cardiomyocyte and therefore cardiac contraction and relaxation are intimately linked to metabolic capacity. Normally, mitochondrial ATP production is generally matched by cellular ATP consumption, which is largely accounted for by myosin ATPase and SERCA. The Creatine Phosphate shuttle ensures high-energy phosphate delivery from mitochondria to cellular ATPases as well as acting as a buffer for ATP during rapid increases in consumption/demand that accompany increased workload e.g., during exercise (Elliott, Smith & Allen, 1994). This buffer function allows for the lag in the response of aerobic glycolysis to match ATP production with demand and thus prevents the fall in local ATP concentration; however this occurs at the expense of reduced cytoplasmic creatine phosphate (CrP) and raised inorganic phosphate levels (Elliott, Smith & Allen, 1994). T2D has been linked to cardiac mitochondrial dysfunction that manifests as lowered intracellular CrP levels in T2D hearts (Bashir, Coggan & Gropler, 2015). This adaptation does not appear to reduce glycolytic flux but lowered CrP/ATP ratios are known to reduce contractile force and this metabolic mechanism may be an important component of DCM.

As will be explained and explored further below, there is strong evidence that pathological metabolic disturbances of the heart are the fundamental event initiating DCM (Lai et al., 2014; Ritterhoff & Tian, 2017; Karwi et al., 2018; Tan et al., 2020). Indeed, many would argue that metabolic perturbations precede functional changes in many examples of cardiac malfunction. A striking example is provided by recent work showing that levels of the mitochondrial pyruvate carrier mediates pathological cardiac hypertrophy in human heart, and mouse knockout of cardiomyocyte mitochondrial pyruvate carriers resulted in cardiac hypertrophy and reduced survival (Fernandez-Caggiano et al., 2020). Hence, we focus the remainder of this review on cardiac metabolism and substrate use as our thesis is that alterations in metabolism and substrate use may be key aetiological factors in the development of DCM.

The role of sugar and fat in a healthy heart

Cardiac metabolism is classically characterised by the predominant use of fatty acids (FA) as a metabolic substrate (Ritterhoff & Tian, 2017; Kerr, Dodd & Heather, 2017; Abel, 2018). These are primarily transported into the cell via the transporter CD36 (Son et al., 2018), and subsequently into mitochondria via carnitine palmitoyltransferase (CPT-1) where they undergo beta-oxidation (Brown et al., 1995; Kim & Dyck, 2016) (see Fig. 2). Fat is an excellent fuel source when considering the molecules of ATP generated per gram, thus making it particularly suitable for powering the indefinite contractile activity of the heart. Under normal conditions there is limited cardiac storage of fat because—as is the case for most organs—excessive accumulation of triglycerides and associated metabolites from incomplete FA oxidation is associated with lipotoxicity (D’Souza, Nzirorera & Kienesberger, 2016). It is well known that increasing fatty acid availability to the heart results in a marked inhibition of glucose oxidation via the glucose/fatty acid cycle (Shipp, Opie & Challoner, 1961; Randle et al., 1963; Hue & Taegtmeyer, 2009). However other additional mechanisms may be at play which are discussed further below.

Figure 2 Trafficking of GLUT4 and CD36 in cardiomyocytes.

In the well-perfused healthy heart >95% of ATP is produced by oxidative phosphorylation and 60–90% of this is derived from metabolism of fatty acids which is the predominantly available substrate. The extraordinary high energetic demand of the heart is underpinned by an ability to adjust substrate preference to match the energetic demand with the levels of prevailing substrate in the circulation. The ability to ‘shift’ substrate preference is beneficial (Ritterhoff & Tian, 2017). For both fatty acids and glucose, flux analysis supports the contention that it is the delivery across the plasma membrane (PM) that controls the flux through the respective metabolic pathways. CD36 is a multifunctional protein which mediates ~70% of the uptake of fatty acids in cardiomyocytes. GLUT4 is the predominant glucose transporter in cardiomyocytes (Abel, 2004; Luiken et al., 2020). Both recycle between intracellular stores and the PM. In response to insulin or contraction, PM levels of CD36 and GLUT4 increase. (Note that different populations of intracellular GLUT4 are mobilised by insulin or contraction, but this is not shown here for simplicity). Once inside the cell, fatty acids are rapidly converted into fatty acyl CoA by fatty acyl CoA synthase which are then substrates for β-oxidation in mitochondria. Similarly, glucose is rapidly converted to glucose-6-phosphate and metabolised to pyruvate in the cytosol and then enters the TCA cycle. Reduced expression of the mitochondrial pyruvate carrier protein is associated with cardiac hypertrophy in human heart (Fernandez-Caggiano et al., 2020). Hence, cardiomyocyte metabolism is considered to be highly adaptive and tightly regulated by hormonal and contraction signals, coupling fuel use with available substrate and metabolic need.

Whilst, fat is undoubtedly an important fuel source for the heart, the ability to rapidly switch between several metabolic substrates is essential to meet ATP demands in the face of varying environmental conditions. Therefore, cardiac metabolism is actually best characterised as being FA predominant but highly flexible (Ritterhoff & Tian, 2017). The second major cardiac metabolic substrate is glucose, which provides a more rapid source of ATP at a lower oxygen cost (Abel, 2004; Doenst, Nguyen & Abel, 2013). Although fatty acids are an abundant energy source for the heart, the ATP produced per O consumed is less efficient than for glucose (Lopaschuk et al., 2010; Lopaschuk & Ussher, 2016). Despite glucose metabolism providing less ATP for the heart daily, defects in this may be of critical importance to DCM (Fig. 2). Hence, it is important that the role of glucose (both metabolism and transport) is carefully considered.

The facilitated diffusion of glucose into target cells can occur via several isoforms of the GLUT transporter family. Comprising 14 members, coded for by the SLC2 genes and at around 500 amino acids in length, each isoform is primarily distinguished by its substrate specificity (some transport other hexoses such as fructose), rate of activity and tissue localisation (Mueckler & Thorens, 2013). In skeletal muscle and the heart, the two key transporters are GLUT1 and GLUT4. During foetal development glucose is the predominant substrate and GLUT1 is abundantly expressed in order to facilitate the sustained rapid uptake required to support physiological development at this stage (Santalucía et al., 1992). Postnatally, upregulation of GLUT4 and reduced GLUT1 content are observed, in association with levels of the thyroid hormone T3 (Castelló et al., 1994). Despite this, GLUT1 continues to strongly influence basal cardiac glucose uptake (Kraegen et al., 1993). GLUT1 is implicated in the pathogenesis of diabetes, as reduced expression is associated with decreased basal skeletal muscle glucose uptake in human diabetic patients. (Ciaraldi et al., 2005). However, by far the most important transporter in cardiac glucose metabolism, overall function and disease is GLUT4.

GLUT4 mediates the regulation of glucose uptake by insulin sensitive tissues – predominantly muscle and fat. Under resting conditions, GLUT4 is sequestered intracellularly in specialised GLUT4 containing insulin responsive vesicles (IRVs; see Fig. 2) (Gould, Brodsky & Bryant, 2020). Post-prandial elevations in blood glucose concentration are sensed and then responded to by the pancreas through increased release of insulin. Insulin binds to its receptor on myocytes and adipocytes and in turn activates PI3K-Akt dependent and APS dependent pathways (Leto & Saltiel, 2012). These signalling networks ultimately result in the activation of effector proteins that facilitate IRV trafficking to the plasma membrane (PM; Fig. 2). These effectors include the Myo1C motor protein that powers GLUT4 translocation, the exocyst complex that tethers IRVs at the PM and the SNARE proteins that facilitate the fusion of these opposing membranes (Gould, Brodsky & Bryant, 2020). GLUT4 is unique in this mode of regulation.

GLUT4 is essential in matching glucose supply and demand in the tissues where it is needed most, and, therefore, also in the regulation of circulating blood glucose levels. The critical role of this protein is underscored by findings that muscular insulin sensitivity is strongly correlated to GLUT4 protein content (Kraegen et al., 1993), impaired insulin stimulated GLUT4 PM trafficking in the skeletal muscle of T2D is the primary defect defining this disease (Garvey et al., 1998; Gould, Brodsky & Bryant, 2020) and that GLUT4 protein overexpression preserves systemic insulin sensitivity in a commonly employed mouse model of diabetes (Atkinson et al., 2013).

A second potent stimulus to enhance skeletal and cardiac muscle glucose uptake is contraction, i.e., exercise (Richter & Hargreaves, 2013; Richter, 2020). As outlined previously, excitation contraction coupling is an energy intensive process and requires increased oxidation of glucose to maintain a rapid supply of ATP under higher external workloads. Whilst this also requires increased cell surface GLUT4 levels (Fig. 2), aspects of contraction mediated GLUT4 trafficking appear distinct (Wojtaszewski et al., 2000; Richter & Hargreaves, 2013; McConell et al., 2020). For example, the stimuli initiating this process are multifaceted and include a change in intracellular energy status (ATP:ADP ratio) leading to AMPK activation and mechanical stress sensitive mechanisms activating the GTPase Rac1, which controls the remodelling of the cytoskeleton required for vesicle translocation (O’Neill et al., 2011; Luiken, Glatz & Neumann, 2015; Sylow et al., 2017). Further, there is evidence that increased PM GLUT4 levels in the heart are facilitated by distinct SNARE proteins when stimulated by contraction versus insulin (Schwenk et al., 2010) consistent with the notion that insulin and exercise recruit distinct intracellular pools of GLUT4 (Richter & Hargreaves, 2013). These observations in part explain why contraction mediated glucose uptake is unaffected by insulin resistance (Wheatley et al., 2004).

Hence, we posit that GLUT4 provides a crossover between metabolic and contractile regulation of skeletal/cardiac metabolism. This is particularly interesting because DCM is a disease of cardiac contractile deficit, with potential metabolic origins. As noted above, mitochondrial dysfunction in T2D heart has been reported which manifests as lowered CrP levels (Bashir, Coggan & Gropler, 2015). It has been proposed that cardiac AMP-activated protein kinase (AMPK) is regulated by creatine phosphate:creatine ratios in a similar way to ATP:AMP in skeletal muscle–therefore when creatine phosphate levels are low, AMPK is activated (Ponticos et al., 1998). AMPK activation in the heart increases glucose utilisation through increased GLUT4 (Glatz et al., 2020)and stimulation of phosphofructokinase (Lefebvre et al., 1996), arguing that increased CrP would attenuate glycolysis. Furthermore AMPK activators increase creatine uptake into cardiomyocytes (Darrabie et al., 2011; Santacruz et al., 2017), which would indicate that when glucose metabolism is low and ATP synthesis is low, increased creatine uptake improves the likelihood of storing ATP as CrP. Such data would suggest that glucose metabolism can influence creatine:phosphocreatine levels in cardiomyocytes (Ritterhoff & Tian, 2017; Glatz et al., 2020) and further reveal the metabolic importance of glucose transport in cardiac metabolism.

Metabolic characterisation of the diabetic heart–linking metabolism to cardiac function

Magnetic resonance (MR) imaging has been used to demonstrate the presence of cardiac diastolic dysfunction in asymptomatic diabetic individuals versus control subjects, despite normal systolic function and left ventricular mass (Diamant et al., 2003). MR spectroscopy revealed a significant association of this impairment with a reduction in the PCr:ATP ratio, indicative of limitations in metabolic function (Abdurrachim & Prompers, 2018). While the association between PCr:ATP and cardiac function has been questioned (Cao et al., 2020), these findings were replicated in a separate study, which correlated this reduction in PCr:ATP with circulating free fatty acid (FFA) concentration (Scheuermann-Freestone et al., 2003).

Increased FFA concentrations are expected in diabetic patients because obesity is one of the major risk factors for this disease. Accordingly, due to an excessive FFA availability that exceeds oxidation requirements or perhaps even capabilities, a key early marker of the diabetic heart is increased triglyceride accumulation (McGavock et al., 2007). In a cohort of middle-aged diabetic men, increased cardiac triglyceride content measured via MR spectroscopy was significantly associated with an impaired cardiac E/A ratio (a marker of diastolic function) (Rijzewijk et al., 2008). This association (independent of other relevant variables) has also been demonstrated in non-diabetic groups, such as obese insulin-resistant women and otherwise healthy aging men (van der Meer et al., 2008; Utz et al., 2011). Furthermore, a reduction in myocardial triglyceride content in response to a weight loss intervention was associated with improved cardiac diastolic function in diabetic individuals (Hammer et al., 2008). Taken together, this suggests that accumulation of FFA in the heart may be a critical factor that impairs metabolic ATP production and is a crucial instigator of DCM.

It is possible that the toxic metabolic effects of cardiac FFA accumulation are indirect. Skeletal muscle insulin resistance is the hallmark of diabetes, however there is also an abundance of evidence demonstrating myocardial insulin resistance and reduced glucose uptake in human diabetic patients (Iozzo et al., 2002a; Dutka et al., 2006; Hu et al., 2018). This is generally demonstrated using positron emission tomography imaging of a radiolabelled glucose analogue that is infused during a euglycemic-hyperinsulinemic clamp. This reduction in cardiac glucose uptake has been demonstrated independent of any limitation in cardiac blood flow, although there is evidence that relative regional perfusion may be altered (Iozzo et al., 2002b). Reduced glucose uptake could have significant consequences for cardiac metabolism and therefore performance due to the aforementioned rapid flux and oxygen efficient generation of ATP via glucose metabolism, particularly in the presence of subclinical cardiovascular stress (likely in patients with diabetes).

As will be explored below, the exact cause of insulin resistance is unclear but may be related to the accumulation of toxic lipid metabolites such as diacylglycerol (DAG), consequent to excessive FFA uptake. Therefore it could be that insulin resistance, at least in part, explains the association of increased cardiac triglyceride content with impaired diastolic function. Consistent with this idea, Rijzewijk et al. demonstrated that in T2D patients, increased cardiac FFA uptake and metabolism coincide with impaired glucose uptake and a reduced E/A ratio (Rijzewijk et al., 2009). Data from animal models confirms that manipulation of circulating FFA levels strongly influences both myocardial triglyceride accumulation and glucose uptake, and also that upregulation of CD36 localisation at the cell surface may facilitate this enhanced FFA entry (Coort et al., 2004; Ouwens et al., 2007; Guzzardi et al., 2014). Interestingly, thiazolidinediones such as pioglitazone have been shown to increase cardiac glucose uptake and improve cardiac diastolic function in diabetic patients, with no reported effect upon FFA uptake (von Bibra et al., 2008; van der Meer et al., 2009). This could imply that restoring insulin stimulated GLUT4 mediated glucose uptake is of primary importance in the diabetic heart.

The evidence discussed so far, predominantly studying asymptomatic T2D individuals, allows us to create logical associations and derive an evidence-based theory regarding the early pathological mechanisms driving DCM. Excessive FFA uptake and therefore accumulation may lead to insulin resistance and starve the heart of the glucose that it needs to rapidly and efficiently generate the ATP that powers excitation contraction coupling. The relatively smaller contribution of glucose to total cardiac metabolism under normal conditions does not diminish its functional importance. However, it is important not to overstate the strength of evidence that we currently have. Human based studies in this field will always face issues with confounding variables (e.g., patient characteristics) and the depth to which investigative techniques can probe subcellular cardiac physiology. Therefore, we must take the principles derived from this research and apply them to credible animal models that represent the human condition as closely as possible. This allows us to perform more invasive manipulative studies and explore these proposed mechanisms further, in the ultimate pursuit of identifying key therapeutic targets.

We suggest three mechanisms that may link impaired metabolism to diastolic dysfunction (Fig. 3).

Figure 3 Possible metabolic abnormalities leading to diastolic dysfunction.

Three possible mechanisms are discussed which may link impaired cardiomyocyte energy metabolism to diastolic dysfunction. These include changes in the balance of fuel use. Increased fatty acid transport and fatty acid cytosolic levels are associated with over-feeding and may lead to insulin resistance and starve the heart of the glucose that it needs to rapidly and efficiently generate the ATP that powers excitation contraction coupling. A high fat diet is known to increase cell surface CD36 levels and reduce levels of GLUT4 at the PM, giving rise to increased fatty acid metabolism and the accumulation of toxic lipid metabolites such as ceramide [1]. Impaired insulin signalling has also been posited as underlying diabetic cardiomyopathy [2]; ceramide is also known to inhibit insulin signalling pathways (see text). Although a relatively small contributor to total cardiac metabolism in the healthy heart (Ingwall, 2009; Ritterhoff & Tian, 2017), in this review, we focus on reduced glucose uptake and decreased levels of GLUT4 [3] as a key underpinning mechanism in diabetic cardiomyopathy and consider whether restoration of GLUT4 levels may be a useful therapeutic intervention.

The first mechanism may involve a direct lipotoxic effect of FFA. Pharmacological inhibition of FA oxidation primarily through reduced CPT-1 mediated mitochondrial uptake was found to increase SERCA expression and ATP availability and ultimately improve cardiac function in diabetic rat hearts (Sharma et al., 2008). However, studies like this are difficult to interpret because a corresponding increase in glucose oxidation was also recorded.

Generally, lipotoxicity is thought to be mediated through FFA derivatives such as DAG or ceramide, which accumulate in diabetic muscle (Itani et al., 2002; Adams et al., 2004). Therefore, it is useful to probe the possible roles of these molecules in DCM. Accordingly, high fat diet fed rodent models of DCM exhibit impaired cardiac efficiency, which can be ameliorated via pharmacological or genetic inhibition of the mitochondrial uncoupling protein UCP3 (Cole et al., 2011; Boudina et al., 2012). Furthermore, mitochondrial uncoupling was directly demonstrated in mitochondrial preparations isolated from db/db mouse hearts (Boudina et al., 2007). UCPs facilitate the movement of protons across the inner mitochondrial membrane without activating ATP synthase, and its upregulation in the diabetic rat heart was found to be regulated by protein kinase C (Arikawa et al., 2007), a key downstream target of DAG. The potential translational significance of this link is enhanced by the observation that in non-diabetic human patients undergoing a cardiac procedure, increased circulating levels of FFA were associated with increased cardiac UCP2 and UCP3 expression and decreased expression of GLUT4, driving energy deficiency (Murray et al., 2004). Murray et al propose that increased mitochondrial UCP expression drives lowering myocardial energy as a result of collapsed proton-gradients across the mitochondria, explaining why cardiac PCr/ATP ratio is negatively correlated with plasma free fatty acids. This is exacerbated by reduced GLUT4 levels, creating an energy deficient myocardium which is highly likely to contribute towards diastolic dysfunction. The impact of GLUT4 levels on PCr/ATP ratio has yet to be defined in cardiac tissue, but transgenic animals offer a potentially useful means to investigate this question.

A second mechanism could be via impaired signalling pathways from the insulin receptor which may link insulin resistance in DCM with impaired cardiomyocyte function. Impaired activation of proximal insulin signalling cascades are consistently reported in diabetic animal models (Ouwens et al., 2005; Deng et al., 2007). Cardiomyocyte specific knock out of the insulin receptor and therefore an inability to activate downstream signalling cascades in mice was shown to result in multiple metabolic impairments, including increased mitochondrial uncoupling and reduced availability of multiple metabolic proteins (Boudina et al., 2009). Additionally, a novel physical interaction between Insulin Receptor Substrate-1 and SERCA proteins has been established, with decreased association in the diabetic rat heart noted suggesting a possible functional link between the most proximal stages of insulin signalling and ECC (Algenstaedt et al., 1997). However, it is unclear how translationally relevant these findings are to the human disease setting. Cardiac biopsies from human diabetic patients revealed an unexpected increase in activation of IRS-1 and PI3K, suggesting that limitations in GLUT4 trafficking in the diabetic heart may not be a consequence of deficits in insulin signal transduction (Cook et al., 2010).

What is clear is that the diabetic myocardium displays IR and impaired insulin action, and that this is fundamentally linked to reduced PM GLUT4 expression (Cook et al., 2010). Therefore, the third and final proposed mechanism linking impaired metabolism and diastolic dysfunction is reduced GLUT4 mediated glucose uptake, which is critical to cardiac function for several reasons already outlined. This is an intriguing theory as it pinpoints a single protein as having a critical disease modifying role. Interestingly, cardiomyocyte specific knock out of GLUT4 in mice induces a similar phenotype to DCM, with cellular hypertrophy accompanied by an impairment in contractility and relaxation that is linked to altered calcium handling and at least partially explained by a significant reduction in ryanodine receptor and SERCA expression (Domenighetti et al., 2010). Furthermore, ex vivo study of perfused db/db mouse hearts revealed characteristic impairments in cardiac contractility and altered substrate utilisation (in favour of FA), all of which was normalised by 4- to 6-fold transgenic overexpression of GLUT4 (Belke et al., 2000). A follow up study with the same model provided similar findings when utilising in vivo echocardiography and demonstrated normalisation of parameters such as the E/A ratio in db/db mice with overexpression of GLUT4 (Semeniuk, Kryski & Severson, 2002). In further support of this argument, perfusion of isolated db/db hearts with a high insulin and high glucose solution increased the efficiency of the diabetic myocardium, and improved recovery post ischemic injury (Hafstad et al., 2007).

Metabolic perturbations are a driving factor in the early pathogenesis of DCM. There is credible evidence citing both increased abundance of FA intermediates and impairments of insulin signalling as important defects, both of which are valuable therapeutic targets. However, considering insight from both human patients and animal models, the major focus of the rest of this review is to consider that restoring cardiac GLUT4-mediated glucose uptake could be of sufficient power to reverse the onset and progression of DCM. This could be a key target for novel therapeutic strategies.

How to develop novel treatment interventions?

Novel treatment strategies that target the cause of DCM are urgently required. The generally applied framework when developing a therapeutic intervention is to identify biochemical pathways relevant to the pathophysiological event of interest, in order to target key proteins for pharmacological activation/inhibition. This is challenging for DCM, where multiple mechanisms are of relevance at different stages of disease progression.

As outlined, there is strong evidence that impaired insulin stimulated GLUT4 trafficking and therefore glucose uptake are critical early events in DCM, making insulin resistance a priority target. However, numerous factors may contribute towards this, creating a need for multiple points of intervention. For example, pro-inflammatory adipocytokines (IL-6,TNF-α) from enlarged, inflamed adipose tissue and myocardial accumulation of DAG from increased FA uptake have both been linked to inhibition of proximal insulin signalling intermediates (Plomgaard et al., 2005; Erion & Shulman, 2010). Additionally, altered SNARE protein expression could indicate disruption of the effector proteins that mechanically regulate GLUT4 translocation (Schlaepfer et al., 2003; Lancha et al., 2015). When combined with multiple secondary factors implicated in the pathogenesis of DCM such as advanced glycaemic end-products, reactive oxygen species and local inflammation (Murtaza et al., 2019) it becomes challenging to decide where to begin.

Further complications for drug development include a reliance on the animal models that have been invaluable in furthering our understanding of cardiac physiology, yet fall short of replicating human disease (Bugger & Abel, 2014). Therapeutic intervention has also been hampered by the common issue among pharmacological agents under development of unacceptable off target cardiotoxicity.

While there are numerous examples of successful drug development for various diseases, in the specific context of DCM there may be an alternative treatment that could be implemented either alongside, or in place of, conventional pharmacological strategies. Ideally this treatment would be capable of restoring cardiac GLUT4 mediated glucose uptake, whilst also directly positively impacting other identified pathophysiological mechanisms (not just indirectly through effects upon GLUT4). Preferably it would also be suitable for testing directly upon humans in order to speed up the development process and remove any need for animal research. This treatment could potentially be adapted from one of the longest standing, easy to implement, cost-effective interventions at our disposal-exercise.

Exercise—a key tool in the battle against cardiovascular disease and diabetes

It is well established that physical activity is fundamental to metabolic and cardiovascular health, dating back to the pioneering study of the incidence of heart disease in London bus drivers and conductors (Morris & Raffle, 1954). Since then, research has continually demonstrated the beneficial effect of exercise upon a range of physiological functions, in both health and disease (see Fig. 4) (Thyfault & Bergouignan, 2020; Pillon et al., 2021).

Figure 4 Key mechanisms through which exercise may improve cardiac function in DCM.

Shown are several mechanisms through which exercise training may alleviate cardiac dysfunction associated with DCM. Firstly, exercise may normalise cardiac metabolism, which could directly enhance cardiac contractile function through increasing ATP availability. Similarly, exercise has been shown to alleviate impairments in cardiomyocyte calcium handling, which could directly enhance contractile function through more efficient use of ATP. It is unclear if these adaptive mechanisms occur independently or if there is a functional link. Finally, general global benefits of exercise training such as weight loss could reduce the burden on the diabetic heart, thus improving function. Please note that we have limited our discussion to what we consider to be key potential mechanisms; see main text for further details.

Overall cardiovascular ‘fitness’ is often represented by the maximum amount of oxygen an individual can take up and utilise (VO2max). This is dependent upon the oxygen carrying capacity of the blood, the rate of oxygen diffusion across peripheral and pulmonary capillaries (influenced by capillary density) and the oxygen utilisation capacity of peripheral tissues (mostly skeletal muscle); however, it is primarily representative of cardiac pump capacity (Bassett & Howley, 2000). The ease of collecting and analysing expired gas samples during a treadmill or stationary bike ramp protocol versus advanced cardiac imaging techniques also makes this parameter relatively accessible for cardiovascular research.

Whilst additional factors, such as running economy and anaerobic threshold, are critical determinants in separating athletes during a race, in a general population VO2max is an excellent predictor of endurance capacity (Larsen & Sheel, 2015) and the gold standard for measuring aerobic power. VO2max is also a strong predictor of an individual’s risk of developing both insulin resistance and cardiovascular disease (Laukkanen et al., 2007; Leite et al., 2009) and also the risk of mortality in heart failure patients (Cohn et al., 1993).

Research has repeatedly demonstrated the capacity of exercise training to improve VO2max in healthy individuals, to the extent that now the effect of relevant variables such as work intensity (Milanović, Sporiš & Weston, 2015) or session structure (Sindiani et al., 2017) to maximise this benefit are now being investigated. Critically, high intensity exercise has also been shown to effectively increase VO2max in patients with coronary artery disease, heart failure or post-MI (Wisløff et al., 2002, 2007; Moholdt et al., 2012; Liou et al., 2016), demonstrating the therapeutic potential of exercise and the plasticity of the cardiovascular system even when operating at reduced capacity.

The ability of exercise to induce chronic adaptations in cardiac structure and function has also been shown via techniques such as transthoracic echocardiography. The concept of an ‘athlete’s heart’ is well defined; prolonged adherence to exercise training results in hypertrophy of the left ventricular wall (Trachsel et al., 2018). Endurance training may also increase the internal diameter of the left ventricular chamber in response to chronic volume overload (eccentric hypertrophy), whereas more resistance-based training does not (concentric hypertrophy), although few athletes adhere to only one type of training (Fagard, 1992). Whilst hypertrophy is associated with negative outcomes in the context of cardiac disease, with exercise training this is a beneficial adaptive response that aids contractile output and relies upon distinct signalling pathways, in particular activation of IGF-1-receptors and PI3K signalling (Neri Serneri et al., 2001; McMullen et al., 2003). Additionally, imaging of the heart also reveals that exercise has the capacity to greatly enhance cardiac diastolic function, as reflected in the velocity of tissue movement and blood flow, in both healthy individuals and heart failure patients (Levy et al., 1993; Pearson, Mungovan & Smart, 2017).

Exercise training can also elicit metabolic adaptations, including in tissues other than skeletal muscles (Thyfault & Bergouignan, 2020). In healthy individuals a key adaptation to endurance training is increased capacity to oxidise fat as a fuel source during acute bouts of exercise, even at workloads of higher intensity (Klein, Coyle & Wolfe, 1994; Coggan et al., 2000). This preserves limited carbohydrate stores and is attributed to increased skeletal muscle mitochondrial content and therefore capacity (Jacobs et al., 2013; Scalzo et al., 2014). For individuals with either T1D or T2D exercise training is even more impactful. Evidence demonstrates that aerobic, resistance or combined exercise training may improve glycaemic control in diabetic patients, whether measured by blood glucose concentration or circulating HbA(1c) (Maiorana et al., 2002; Sigal et al., 2007; Tomas-Carus et al., 2016; Röhling et al., 2016), the latter of which has been shown to be an excellent predictor of mortality risk (Khaw et al., 2001). Similar to healthy individuals, this may be in part independent of exercise induced weight loss (Boulé et al., 2001), and instead attributable to improved insulin sensitivity (glucose uptake) and enhanced mitochondrial function (O’Gorman et al., 2006; Meex et al., 2010; Little et al., 2011).

Exercise may also specifically halt the progression of DCM

Short term (typically 12 weeks) exercise-based interventions may enhance impaired diastolic capacity in patients with T2D, with higher intensity exercise typically producing superior results and, in some cases, restoring function to control values (Brassard et al., 2007; Hollekim-Strand et al., 2014; Cugusi et al., 2015; Cassidy et al., 2016). While there are studies where exercise was found to have no beneficial effect (Loimaala et al., 2007), longer-term follow-ups (1 and 3 years) indicated that this may be linked to either poor adherence or insufficient exercise intensity (Hordern et al., 2009; Hare et al., 2011). A recent meta-analysis also concluded that exercise can exert beneficial effects on cardiac diastolic function in diabetic patients (Verboven et al., 2019). The extent of those effects are dependent upon several factors, including the measurement tool and parameters of choice, and the nature of the implemented training.

Exercise has been proposed to elicit beneficial effects through multiple mechanisms, as recently reviewed (Hafstad, Boardman & Aasum, 2015; Seo et al., 2019) and indicated in Fig. 4. The extent to which we can probe human tissue for further insights is limited; however, when focussing on key functional parameters from rodent models, it appears that normalising calcium handling is critical. In db/db mice, an interval-based exercise programme returned systolic and diastolic function of the whole heart in vivo and in isolated cardiomyocytes to levels recorded in wild type controls, primarily attributable to normalisation of the calcium transient (Stølen et al., 2009). This in turn was linked to restored sarcoplasmic reticulum calcium loading through enhanced expression and activity of SERCA, and preservation of the t-tubule network. Whilst specific details vary across models or studies, a consistent finding in this field is exercise induced restoration of cardiomyocyte calcium handling dynamics linked to enhanced SERCA activity (Mishra et al., 2011; Epp et al., 2013).

Overall, it is clear that exercise training can enhance diastolic cardiac function in diabetic patients, most likely through a multi-faceted mechanism. There is good evidence that altered substrate utilisation and subsequent mitochondrial dysfunction and limitations in ATP availability are linked to cardiac dysfunction in DCM. Therefore, it is perhaps unsurprising that exercise-based interventions have been shown to enhance mitochondrial biogenesis, structure and function (ATP availability) and restore myocardial glucose utilisation (Fig. 4) (Searls et al., 2004; Broderick, Poirier & Gillis, 2005; Wang et al., 2015). Moreover, exercise-based interventions have had positive effects on cardiac metabolism in humans and rodents (Shao & Tian, 2015; Gibb & Hill, 2018). Whilst there is no direct evidence conclusively demonstrating that these adaptations are the primary instigator of improved calcium handling, it is a reasonable theory that should be investigated by future work.

Notably, one study reported that cardiac lipid content was unaltered by an exercise intervention (Schrauwen-Hinderling et al., 2011). This is arguably unsurprising as trained individuals commonly have elevated levels of intramuscular lipids that correlate with high levels of insulin sensitivity (the ‘athlete’s paradox’). Exercise may alleviate lipotoxicity through a reduction in toxic lipid intermediates (e.g., ceramide), improved mitochondrial efficiency and increased lipid turnover, as opposed to simply reducing total cellular lipid content. This underscores the multi-faceted impact of exercise training upon metabolic function.

Can we specifically target GLUT4 with exercise?

The fundamental principle of exercise training is that if we impose specific demands upon a physiological system then with subsequent rest and recovery a specific adaptation process will occur that will enhance performance in response to future stress. Despite this, the use of generic strength and/or endurance exercise interventions in scientific research is widespread. There are sound experimental reasons for this approach e.g., to standardise workloads across subjects, yet the limitations are rarely acknowledged. Exercise is often referred to and discussed as a singular entity, rather than as an adaptable process allowing a diverse range of stimuli to be implemented. Therefore, prior work investigating the impact of exercise on DCM may have barely scratched the surface of what can be achieved. Previous studies with one standardised intervention, often with limited overall loading, are likely to have produced sub-optimal improvements.

In theory it should be possible to design an exercise-based intervention that specifically targets maximal upregulation of insulin-stimulated GLUT4 mediated glucose uptake in patients with DCM (Fig. 4). Rather than relying on the generally beneficial effects of exercise, training would be tailored to target alleviation of cardiac insulin resistance–one of the fundamental events in the early pathogenesis of DCM. In this regard it may be useful to consider strategies known to enhance GLUT4 expression in skeletal muscle (Richter & Hargreaves, 2013), as similar regulatory networks may also operate in heart tissue.

Prior work with healthy rats demonstrates that exercise regimes can increase cardiac GLUT4 protein expression (Palabiyik et al., 2016; Schaun et al., 2017). Endurance running training has also been found to prevent large diabetes-induced decreases in rat myocardial GLUT4 expression, demonstrating therapeutic potential (Hall, Sexton & Stanley, 1995; Osborn et al., 1997). Recent studies have also revealed exercise-induced beneficial effects on cardiac metabolism (Gibb & Hill, 2018) and in human muscle (Wojtaszewski et al., 2000; Richter & Hargreaves, 2013; Richter, 2020) and parallel changes in skeletal muscle and cardiac GLUT4 levels in response to changes in plasma free fatty acid levels (Murray et al., 2004). Collectively, this work supports the notion that exercise can impact GLUT4 in the diabetic heart but provides no information regarding optimal conditions for human patients.

Separately, multiple studies have demonstrated the capacity of exercise interventions (of varying design and duration) to induce increases in skeletal muscle and adipose tissue GLUT4 protein expression in human diabetics, alongside improved glycaemic control (Dela et al., 1994; Holten et al., 2004; O’Gorman et al., 2006; Hussey et al., 2011; Little et al., 2011). Comparison of trained and sedentary individuals has positively correlated increased insulin mediated phosphorylation of PI3K and glucose uptake in athletes (Kirwan et al., 2000); however, evidence from the obese Zucker rat diabetic model indicates that an increase in GLUT4 protein expression is the primary mechanism that can drive alleviation of skeletal muscle insulin resistance (Etgen et al., 1997; Ivy, 2004). Furthermore, in one study exercise increased whole body insulin mediated glucose disposal despite unchanged activation of key proximal insulin signalling intermediates in skeletal muscle (O’Gorman et al., 2006). This further supports the notion that increasing GLUT4 protein expression in the heart should be the primary target of interventions to alleviate cardiac insulin resistance in the context of DCM.

Exercise enhances GLUT4 expression across all major insulin target tissues. This alone may be sufficient to alleviate tissue specific insulin resistance, as suggested by prior work overexpressing GLUT4 in diabetic mouse models (Belke et al., 2000; Semeniuk, Kryski & Severson, 2002). Moving forwards, the most beneficial stimulus to induce the desired adaptations should be identified. Arguably most importantly, the optimal work intensity should be determined. It could be that high intensity interval training (HIIT) is key in order to achieve the maximal rate of glucose uptake possible, or alternatively that steady state endurance training is preferable in order to maximise the total amount of glucose taken up over the whole exercise session. Interestingly, HIIT has previously been found to be most effective at improving diastolic function in diabetic individuals and also aerobic fitness in other cardiac rehabilitation settings (Hollekim-Strand et al., 2014; Cassidy et al., 2016; Hannan et al., 2018). Similarly, the most effective exercise modality and frequency must be identified. Finally, the programming most appropriate to different stages of disease progression should be established, as clinical manifestation can vary from asymptomatic (early stage) to severe debilitation (late stage).

The optimal conditions would be easier to identify if the mechanisms regulating GLUT4 expression were better understood. Numerous transcription factors have been cited as having a regulatory role over GLUT4 gene transcription, with particular emphasis on MyoD, MEF2 and the Thyroid hormone receptor (Zorzano, Palacín & Gumà, 2005). Accordingly, increased nuclear localisation and DNA binding of MEF2 has been identified in the skeletal muscle of human subjects after an acute exercise bout (McGee et al., 2006). Moving forwards, a comprehensive list of regulatory factors must be detailed, before assessing the impact of key exercise variables upon their activity. Additionally, increased skeletal muscle GLUT4 protein content independent of chronic changes in GLUT4 mRNA were noted in response to a 4-week running programme in high-fat fed mice, suggesting that regulation of the expression of GLUT4 may not be limited to transcriptional mechanisms (Gurley, Griesel & Olson, 2016).

Side effects and limitations of exercise

The aim of this review is not to argue that exercise programmes should replace all pharmacological strategies in the management of DCM. Rather, we seek to highlight the underappreciated and under investigated potential of exercise to act as a first point of intervention. A further benefit of appropriately managed exercise training is that unlike many drugs there is a lack of off target adverse side effects.

Media reporting of exercise-induced sudden cardiac death (particularly in mass-participation sports) can elevate the perceived risk; however, in reality it is rare. Estimates place the risk at between 1:40,000 to 1:80,000, with the majority of cases linked to previously undiscovered cardiomyopathy (de Noronha et al., 2009; Harmon et al., 2014). A ‘J-shaped’ relationship between exercise duration and risk of sudden cardiac death has been described previously; however, any elevation in risk may not be observed until weekly exercise duration exceeds 10 h, which is unlikely to apply to training designed for individuals with DCM. Additionally, evidence linking long term regular strenuous exercise with potentially adverse cardiac remodelling (fibrosis; coronary artery calcification) does not stand up against the clear reduction in major cardiovascular events and mortality observed in moderately and highly physically active individuals (Eijsvogels et al., 2016).

Given the nature of DCM, it is logical to suggest that there could be an elevated risk in this patient population. However, a meta-analysis of randomised control trials including data from over 6,000 post-MI patients demonstrated exercise-based interventions significantly reduced their risk of reinfarction and all-cause mortality (Lawler, Filion & Eisenberg, 2011). Similarly, a meta-analysis of heart failure patients demonstrated a reduced rate of rehospitalisation and improved quality of life in response to exercise rehabilitation (Taylor et al., 2019). Importantly, a study of 4,846 coronary heart disease patients undergoing exercise-based rehabilitation demonstrated extremely low rates of exercise induced sudden cardiac death, even during sessions of higher intensity (Rognmo et al., 2012). Overall, this suggests that properly designed exercise programmes for patients undergoing rehabilitation from various cardiac diseases are safe, and that the benefits accrued greatly outweigh any associated risk.

The biggest limitation of exercise-based interventions is undoubtedly adherence. Understanding reasons for non-completion and how to overcome psychological barriers to adherence are key areas of research (Bock et al., 1997; Aamot et al., 2016). This could mean adapting prescribed workouts to meet individual needs and preferences. Consideration of this should be central to the design of any intervention, as adopting a holistic approach will provide the greatest chance of success. Perhaps this is an early-stage version of personalised medicine!

If properly implemented, any side effects associated with exercise training are more likely to be positive. As stated, one of the key benefits of exercise is that it has the capacity to improve multiple aspects of physiological function. Therefore, even if designing an intervention to enhance cardiac glucose uptake, it is likely that other aspects of patient health would improve. For example, regular exercise combined with caloric restriction will likely result in clinically significant (>5%) weight loss (Swift et al., 2018). Obesity is not only a major risk factor for diabetes, but also for multiple other conditions such as coronary heart disease (Abbasi et al., 2002). These benefits can be accrued at nominal cost, therefore allowing healthcare resources to be focussed where they are needed most. The recently described remission of T2D by dietary modification provides a clear illustration of this (Taylor, 2019), and the impact of this on cardiac function is eagerly awaited. Finally, exercise has been recognised as a powerful tool to alleviate the symptoms of anxiety and depression (Carek, Laibstain & Carek, 2011; Jones et al., 2016). This means that exercise-based interventions could enhance the psychological well-being of DCM patients through both direct and indirect mechanisms.

One size may not fit all

Increased aerobic exercise capacity is a key health-promoting benefit of regular aerobic exercise, but humans vary considerably in their ability to respond to exercise. The ‘exercise capacity’ phenotype has been increasingly recognised as a predictor of health and longevity (Grundy et al., 2012; Kokkinos et al., 2014), raising the interesting question of what might limit the capacity of individuals to respond–might there be environmental or genetics factors at work?

Building on evidence that has linked low exercise capacity and hyperglycaemia, MacDonald et al evaluated whether chronic hyperglycaemia–specifically consumption of a ‘western diet’ or the induction of experimental diabetes—impacted exercise-induced improvements in aerobic capacity and skeletal muscle remodelling in rodents. This study revealed that exercise-induced improvements in aerobic capacity and associated skeletal muscle remodelling were impaired in models associated with chronic hyperglycemia (MacDonald et al., 2020). This ground-breaking study further identified a signalling network that is aberrantly activated by aerobic exercise in rodents with hyperglycaemia (and also humans). This involved both glucose-mediated modifications of the extracellular matrix and altered JNK-signalling in muscle (MacDonald et al., 2020).

Somewhat surprisingly, mice with hyperglycaemia exhibited elevated GLUT4 muscle levels and improved glucose tolerance with training. As noted by MacDonald, this hints that ‘metabolic improvements with exercise are dissociated from improvements in exercise capacity’ (MacDonald et al., 2020), which raises the possibility that exercise-induced improvements in glucose metabolism might reduce the burden of low improvements in aerobic capacity. Future work in this area will be of great interest. The increasing incidence of hyperglycaemia may therefore be acting in as yet unappreciated ways to impair metabolic health. Whether such adaptations also underpin cardiac muscle function remain unexplored.

Finally, a recent study from Abel and colleagues noted that that restoring glucose delivery to cardiomyocytes in the context of streptozotocin-induced diabetes could result in acceleration of mitochondrial dysfunction, an effect mediated by elevated O-GlcNAcylation of the transcription factor Sp1 (Wende et al., 2020). They argue this work supports previous studies indicating that myocardial insulin resistance (i.e., decreased glucose uptake) may be cardioprotective (Taegtmeyer et al., 2013) and posit that the reduced glucose utilisation observed in diabetic cardiomyopathy may act as a defence mechanism against glucotoxicity. This latter point needs to be investigated directly, as data from streptozotocin-induced diabetes may not correlate with DCM. It is well established that GLUT4 is the major contributor to glucose uptake in the beating heart and that GLUT4 levels are reduced in DCM, prompting Wende et al assert that ‘increasing or maintaining GLUT4 expression in the heart represents a reasonable approach for preserving myocardial glucose utilisation in the context of diabetes’ (Wende et al., 2020).

Could iPSC-CM help in this field?

Human induced pluripotent stem-cell derived cardiomyocytes (iPSC-CM) are a relatively recently developed technology that have potential applications in the study of cardiac development and disease or in the practical fields of drug toxicity screening and even cardiac regeneration (Del Álamo et al., 2016; Shiba et al., 2016). iPSC-CM are presently considered to represent cardiomyocytes at a foetal stage of development. Whilst they express many cardiac specific genes and proteins and exhibit contractile activity, their structure (lacking t-tubules, poorly organised myofilaments), metabolism (dependence upon glucose) and electrophysiological profile (elevated resting diastolic potential, slow action potential upstroke velocity) require development in order to achieve the level of comparative physiology required for optimal application (Correia et al., 2017; Parikh et al., 2017).

Previously, features consistent with pathological hypertrophy and altered calcium handling were induced in iPSC-CM through exposure to elevated levels of endothelin-1 and cortisol (Drawnel et al., 2014). This was taken as evidence of having developed a physiologically relevant model of DCM. However, a notable limitation of this paper was the absence of any reference to insulin stimulated glucose uptake or related metabolic parameters. To date, only 2 publications have claimed to have induced an insulin resistant phenotype in iPSC-CM, through culturing cells in the presence of elevated levels of palmitate or TNF-alpha and recording reduced insulin stimulated glucose uptake and reduced phosphorylation of Akt (Liu et al., 2017; Chanda et al., 2017). This data is promising; however, it must be noted that at baseline in these studies insulin stimulation increased glucose uptake 1.2–1.5 fold, and in both cases the raw unadjusted data was not presented. This makes these results more difficult to interpret, and may not compare favourably to data from isolated primary cardiomyocytes (Fischer et al., 1997; Mazumder et al., 2004); but see also (Bryant & Gould, 2020).

We investigated the impact of different experimental conditions upon glucose uptake and insulin sensitivity in commercial iPSC-CM (Bowman, Smith & Gould, 2019). Through direct comparison to the highly insulin sensitive murine 3T3-L1 adipocyte cell line and isolated adult cardiomyocytes, it was concluded that under appropriate experimental conditions iPSC-CM do not adequately respond to insulin with regards to their rate of glucose uptake. We have pinpointed very little to no expression of GLUT4 protein as a primary (but not necessarily exclusive) limitation in iPSC-CM in this context (Bowman, Smith & Gould, 2019). In contrast, these cells express an abundance of GLUT1 protein (Bowman, Smith & Gould, 2019). This expression profile is consistent with immature foetal cardiomyocytes (Castelló et al., 1994) and represents another parameter that must be altered in order to advance the maturation status of these cells. It will be interesting to ascertain whether the GLUT1/GLUT4 ratio more closely replicates the adult phenotype in more complex systems, such as cardiac organoids (Mills et al., 2019) or from iPSC-CMs derived using more complex differentiation systems (Parikh et al., 2017; Ronaldson-Bouchard et al., 2018) or grown on different matrices.

The lack of GLUT4 protein and insulin stimulated glucose uptake in this cell type prevents generation of a robust model of DCM with iPSC-CM at this time. However, it does provide a unique opportunity to probe the regulation of cardiac GLUT4 protein expression (and subsequent trafficking) by exercise training. iPSC-CM spontaneously beat in culture; however, their contractile activity can also be regulated through external field-based stimulation or optogenetic control (Lapp et al., 2017). These techniques could facilitate conditioning protocols that provide a stimulus analogous to exercise training. Through varying the frequency, duration, and intensity of this stimulus and observing the effects upon GLUT4 protein expression we could obtain valuable information that can inform optimal strategies for subsequent programmes in humans.

Conclusions

Diabetic cardiomyopathy develops from initial metabolic alterations and leads to progressive functional and structural adaptations that can ultimately result in heart failure. Impaired regulation of the subcellular distribution of GLUT4 has emerged as an important early pathological event. Exercise can counteract a range of pathophysiological mechanisms and consequences of diabetes, and suitable programmes could and therefore should be developed to specifically target GLUT4 alongside current pharmacological strategies. The rapidly developing iPSC-CM technology could be of benefit in this field.

We are most grateful to Tomas Stølen (Norwegian University of Science and Technology and Trondheim University Hospital) for helpful and insightful comments on the manuscript. We also thank the reviewers for helpful and constructive comments.

Additional Information and Declarations

Competing Interests

Author Contributions

Data Availability

Gwyn W. Gould is an Academic Editor for PeerJ. There are no other competing interests.

Peter R.T. Bowman conceived and designed the idea for the review, performed the literature search, analyzed the output, authored or reviewed drafts of the paper, and approved the final draft.

Godfrey L. Smith performed the literature search, analyzed the output, prepared figures and/or tables, authored or reviewed drafts of the paper, and approved the final draft.

Gwyn W. Gould conceived and designed the idea for the review, performed the literature search, analyzed the output, prepared figures and/or tables, authored or reviewed drafts of the paper, and approved the final draft.

The following information was supplied regarding data availability:

This is a review article, no primary data, code or other information was generated during the course of this work or used in the article.

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
