# Peer review of "Run for your life: can exercise be used to effectively target GLUT4 in diabetic cardiac disease?"

_PeerJ, doi:10.7717/peerj.11485_

## Round 0.1 · original submission · Minor Revisions

The reviewers responded positively to your review. They both had suggestions for changes that would improve the manuscript.

Reviewer 1 ·

Basic reporting

Bowman et al provide a comprehensive review of cardiac metabolism and how it related to diabetic cardiomyopathy. This review provides insight into the links between insulin action, metabolism and cardiomyopathy. The Authors also provide an opinion on how exercise may overcome diabetic heart complications. This is an interesting topic and interesting article.

Language – Overall the article is well written.

The review generally covers field with required references. However, the Authors do not mention the recent publications linking heart pyruvate transport into mitochondria with cardiac hypertrophy. Is this relevant to their argument lining glucose metabolism with diabetic heart? https://www.nature.com/articles/s42255-020-00276-5

The review is comprehensive and covers a lot of ground. However, it is also quite long. It would be useful to edit for conciseness if possible.

Experimental design

Survey methodology is clear and appropriate.

Validity of the findings

The Authors clearly outline to context and scope of the review.

Additional comments

1. The Authors make a put together an argument that altered fuel utilisation (lower glc) leads to impaired ATP or CrP for contraction– “Excessive FFA uptake and therefore accumulation may lead to insulin resistance and starve the heart of the glucose that it needs to rapidly and efficiently generate the ATP that powers excitation contraction coupling. The relatively smaller contribution of glucose to total cardiac metabolism under normal conditions does not diminish its functional importance.” This appears to be a key argument, although as the Authors state, the evidence is limited. Should this be include in figure 3?
a. However, the authors also state “that “T2D has been linked to cardiac mitochondrial dysfunction that manifests as lowered intracellular CrP levels in T2D hearts (Bashir, Coggan & Gropler, 100 2015). This adaptation does not appear to reduce glycolytic flux” – these data appear to this point to there being no change in glucose metabolism. Is there any evidence for functional coupling between glucose metabolism (and not FA metabolism) and CrP?
b. If this is not linked directly to glucose metabolism, what are other possible mechanisms by which insulin resistance lowers the CrP concentrations?
2. Line 246 onwards:
a. Increased UCP3 expression is proposed to link FFAs to cardiomyopathy. But the Authors do not directly relate this change back to ATP and CrP – its implied, but a direct statement would be helpful.
b. It might be useful to include a statement that the idea that the ATP/CrP ratio can be linked to GLUT4 has not been tested, but could be tested in KO or TG models.
3. "Can we specifically target GLUT4 with exercise" – Is there any data on how exercise increases GLUT4 expression in heart? How might GLUT4 be stabilised at the protein level in response to exercise? Is this a reasonable target for therapy?
4. The section on exercise is lengthy and might benefit from editing to make sure it’s a concise as possible. In addition, this section is about 50% of the review, but has no summary figure. It would help the reader to have a figure for reference.
5. Exercise interventions are notoriously difficult to maintain – how do the authors propose to overcome this limitation?
6. The discussion of the use of IPCS cells is excellent. Perhaps the Authors could extend very slightly to talk about organoids - https://www.cell.com/cell-stem-cell/pdfExtended/S1934-5909(19)30108-0. Do these 3D cultures have a better GLUT4/GLUT1 expression profile?

Figures:
Figure 2 – the figure suggests that contraction and insulin both increase glucose and fatty acid uptake. How does the heart balance the use of each substrate (text or legend). Is it purely what is available in the circulation? Does the Randle cycle dictate use of FAs over glucose, but both CD36 and GLUT4 are mobilised so that any available substrate is taken up?
Figure 3 – describes the links between altered metabolism and diastolic dysfunction. Changes in ceramide are linked to changes in insulin signalling, so should there be a link between ceramides and insulin signalling or even directly to GLUT4? In addition, while this figure highlights the changes in metabolism that may link to diastolic dysfunction, there is no information on how these link to contraction. Could this be included?

·

Basic reporting

See general comments.

Experimental design

NA

Validity of the findings

NA

Additional comments

Many thanks for your review. My bias is that all pharmacological interventions should be judged versus exercise, so I enjoyed reading this review.
Line 46: Slightly awkward sentence structure.
Line 59-62: As above. Considering separating into 2 sentences?
Line 73: I find the logic in this subsection a little difficult to follow. You’re saying that IR precedes deterioration of cardiac function, therefore DCM is a consequence of diabetes?
Line 140: Period after the reference.
Line 156: Sentence structure is off. Consider commas either side of therefore?
Line 163: needs a comma after exercise
Line 187: missing a reference
Line 198: Period after the reference
Line 220: missing some commas
Line 246: I wonder if you could comment on glucose-fatty acid cycle and how this may affect glucose uptake in the heart. I think this model has been larhgely disregarded by the work of Shulan and others, but I believe it is still a widely held belief.
Line 356: Anaerobic threshold, rather than lactate?
Line 412: Perhaps some comment about the athletes paradox is warranted here. Exercise may not decrease TGs, but it may decrease the lipotoxic intermediates.
Line 416: You draw parallels to insulin signalling in cardiac and skeletal muscle at the beginning of the review. You make the important point that it is difficult to asses the effects of exercise on cardiac tissue. Do you think you can infer this from the effects on skeletal muscle, which have been well investigated?
Line 442: I’m unconvinced re: the argument that there are responders and non-responders to exercise training. And I disagree with the statement that this is a widely-held belief. I would also question the statement that all “individuals can adapt to exercise training, however the required load, optimal programme and degree of adaptation achieved will show inter-individual variability”. In the context of type 2 diabetics who, I do not think it would be unfair to say, are likely well-below average ‘fitness’. For these individuals, any structured exercise is likely to induce favourable, potentially disease modifying adaptations.
Line 461: Again, I think it would be useful to draw parallels with SM here. Presumably, the type of exercise best suited to increase SM GLUT4 translocation would also apply to cardiac muscle?
Line 513: It might be useful to review how drugs may interact with exercise to modulate adaptations. In particular, the effects of metformin may be interesting.
Line 521: 1:40,000? rather than 1:40,00
Line 522: My understanding is this J shaped relationship is only observed when weekly duration exceeds ~>10 hours, so may not be a factor here?
Line 630: gap after reference
Line 598: I find the section on iPSC-CM a little too speculative to warrant inclusion, but it’s not a deal breaker.
Once again, thank you for your efforts in bringing this review together. I look forward to hearing your responses to my recommendations in due course.

---

## Round 0.2 · accepted · Accept

The authors have revised their manuscript and one of the original reviewers is happy with this new version.

·

Basic reporting

See below

Experimental design

See below

Validity of the findings

See below

Additional comments

Many thanks to the authors for their revisions and consideration of the points I raised.